# Smart Devices for Health and Wellness Applied to Tele-Exercise: An Overview of New Trends and Technologies Such as IoT and AI

**DOI:** 10.3390/healthcare11121805

**Published:** 2023-06-20

**Authors:** Antonio Fabbrizio, Alberto Fucarino, Manuela Cantoia, Andrea De Giorgio, Nuno D. Garrido, Enzo Iuliano, Victor Machado Reis, Martina Sausa, José Vilaça-Alves, Giovanna Zimatore, Carlo Baldari, Filippo Macaluso

**Affiliations:** 1Department of Theoretical and Applied Sciences, eCampus University, 22060 Novedrate, Italy; antonio.fabbrizio@uniecampus.it (A.F.); alberto.fucarino@uniecampus.it (A.F.); manuela.cantoia@uniecampus.it (M.C.); andrea.degiorgio@uniecampus.it (A.D.G.); enzo.iuliano@uniecampus.it (E.I.); martina.sausa@studenti.uniecampus.it (M.S.); giovanna.zimatore@uniecampus.it (G.Z.); carlo.baldari@uniecampus.it (C.B.); 2Research Center in Sports Sciences, Health Sciences and Human Development, CIDESD, 5000-801 Vila Real, Portugal; ngarrido@utad.pt (N.D.G.); vmreis@utad.pt (V.M.R.); josevilaca@utad.pt (J.V.-A.); 3Sciences Department, University of Tras-os-Montes & Alto Douro, 5000-801 Vila Real, Portugal

**Keywords:** tele-exercise, physical activity, smart devices, telehealth, healthy lifestyle

## Abstract

This descriptive article explores the use of smart devices for health and wellness in the context of telehealth, highlighting rapidly evolving technologies such as the Internet of Things (IoT) and Artificial Intelligence (AI). Key innovations, benefits, challenges, and opportunities related to the adoption of these technologies are outlined. The article provides a descriptive and accessible approach to understanding the evolution and impact of smart devices in the tele-exercise reality. Nowadays, technological advances provide solutions that were unthinkable just a few years ago. The habits of the general population have also changed over the past few years. Hence, there is a need to investigate this issue and draw the attention of the scientific community to this topic by describing the benefits and challenges associated with each topic. If individuals no longer go to exercise, the exercise must go to their homes instead.

## 1. Introduction

Physical activity is a crucial element in managing or avoiding various illnesses, but there are several obstacles that hinder people from engaging in exercise [1,2]. These obstacles involve limited access to fitness facilities, insufficient time, and the expenses associated with fitness programs [3]. Given the above, thanks to the recent pandemic and the fact that tele-exercise has become more popular, the practice of the latter has increased, above all helping people with different pathologies [4,5]. Moreover, tele-exercise is a constantly evolving field that uses digital technologies to provide remote exercise and training programs [6]. This concept has become increasingly popular in recent years due to the spread of smart health and wellness devices and the growing need to find flexible and convenient solutions for maintaining an active and healthy lifestyle [7].

Tele-exercise offers a wide range of solutions to meet users’ individual needs and preferences, such as the use of mobile fitness apps that offer guided workouts on video, personalized programs, activity and nutrition tracking [8], online workout programs with live instructors, personalized video conferencing, remote health monitoring, wearable devices, and environmental sensors [9]. 

The use of smart devices such as wearables and environmental sensors has made it possible to collect real-time information about users’ physical activities as well as main physiological parameters such as heart rate, blood pressure, blood oxygen saturation, and more [10]. This data is used to personalize training programs and provide accurate and timely feedback to users, enabling them to monitor their progress and achieve their fitness goals more effectively.

Artificial Intelligence (AI) technology has become increasingly important in the field of tele-exercise [6,11], allowing workout programs to be customized to users’ individual needs and abilities [11]. AI can be used to analyze data collected from wearable devices and environmental sensors and provide personalized recommendations and feedback. For example, AI can help identify users’ strengths and areas for improvement by suggesting specific exercises to achieve their fitness goals [12].

Another important aspect of tele-exercise is its ability to provide a personalized and flexible workout experience. Thanks to digital technologies, users can access tailored training programs that fit their needs and lifestyle [13,14]. In addition, tele-exercise enables users to overcome geographical and access barriers, offering the ability to work out anywhere and anytime, with a wide range of workout options to choose from.

The development and adoption of emerging technologies such as the Internet of Things (IoT) and AI have further expanded the potential of this field, offering new opportunities to monitor and improve people’s health and well-being remotely. This review article aims to examine the role of smart devices in tele-exercise, analyzing the main innovations, benefits and challenges associated with these technologies and discussing opportunities for future research in this area.

The main objective of this article is to provide a narrative, descriptive review of new trends and technologies in the field of tele-exercise, as reported in the literature in the last three years, with a focus on the use of smart devices and the integration of IoT and AI. 

A targeted search on PubMed of the term “tele-exercise” associated with several topics yielded results from 2021 to 2023. It is apparent that there is currently a low number of scientific articles (The data are shown in Table 1 below). In addition, tele-exercise has currently been mainly associated with a rehabilitation phase or a forced choice of home exercise. Although there are already dedicated tele-exercise apps on the market, they are still in their early versions. Studies on the benefits that this type of activity can give have not yet been carried out on a massive scale. Thus, attention should be focused on how to develop an ideal app for all possible future tele-exercise users. In the next section, we will describe the world of physical activity carried out via tele-exercise.

Figure 1 illustrates the exponential growth of the number of papers published on the topic of “tele-exercise”. The results indicate a varying number of publications on the topic of “tele-exercise” over the years. It is worth noting that the number of papers appears to have increased in recent years, with a peak in 2022; this is because 2023 is only partially represented. This suggests a growing interest and research activity in the field of tele-exercise. However, it is important to consider that the number of papers alone does not provide a comprehensive view of the quality or impact of the research conducted.

## 2. Emerging Technologies Related to Tele-Exercise

### 2.1. Internet of Things (IoT)

The Internet of Things is a network of connected devices that collect and share data to improve the user experience and offer personalized services. In the context of tele-exercise, IoT devices can include smartwatches, activity trackers, smart scales, and environmental sensors. These devices allow users to monitor various parameters, such as heart rate, sleep quality, and posture, providing real-time feedback and enabling users to optimize their training and lifestyle [46,47,48,49]. Direito et al., through their analytical work, offer a systematic review of the latest developments in emerging technologies for tele-exercise. The authors examine the effectiveness of eHealth technologies in promoting physical activity. The authors analyze a variety of eHealth interventions, including smartphone apps, websites, and wearable devices, assessing their impact on exercise adherence and health-related outcomes [50].

Benefits of IoT:-Effectiveness: the article notes that many of the eHealth technologies reviewed have been shown to be effective in promoting physical activity and improving health-related outcomes;-Broad reach: eHealth solutions have the potential to reach a wide range of users in different geographic areas, overcoming geographic barriers;-Personalization: eHealth interventions can be tailored to individual needs and preferences, helping to improve adherence and motivation;-Real-time monitoring: eHealth technologies can provide immediate feedback and monitor progress over time, which can help users maintain motivation and adapt their physical activity programs.

Challenges of IoT:-Heterogeneity of interventions: the article points out that there is a considerable variety of eHealth interventions, which makes it difficult to determine what the key success factors are and how they can be applied more broadly;-Long-term adherence: many studies included in the systematic review have a limited duration, making it difficult to assess the long-term effectiveness of eHealth interventions and user adherence over time;-Digital inequalities: people with limited access to digital technologies or limited digital skills may not benefit from eHealth interventions to the same extent as more experienced users;-Privacy and data security issues: the collection and storage of users’ personal and sensitive data may raise concerns about privacy and data security, which may affect users’ trust in eHealth technologies.

### 2.2. Artificial Intelligence (AI)

Artificial intelligence is playing an increasingly important role in the field of tele-exercise. By processing large amounts of data collected from IoT devices, AI algorithms can identify patterns and trends that can help improve user health and well-being. Some applications include analyzing workout data to provide personalized suggestions, creating AI-based workout plans, and monitoring health conditions to prevent illness and injury. He et al. examines the practical implementation of artificial intelligence (AI) technologies in medicine, including applications in the field of tele-exercise. Their work highlights how AI can be used to analyze exercise data and provide personalized suggestions, create AI-based exercise plans, and monitor health conditions to prevent injuries [51]. Liu et al. present an overview of personalized fitness systems based on artificial intelligence and wearable devices that can help users maintain a healthy and active lifestyle. They also discuss how AI can be used for workout data analysis, the creation of personalized workout plans, performance evaluation, and user motivation through personalized interaction. In their analysis, the benefits that exercisers receive from AI integration of data appear evident [52]. In recent years, and even more so in recent months, applications that harness the power of AI within the world of tele-exercise have increased significantly. The features that these apps exploit, by AI, are described in the following: (i) Real-time analysis of the movement made by the practitioner subject and the ability to detect errors and provide a suggestion for correction; (ii) Modification of the fitness plan according to the user’s needs and the results set by the user; (iii) Through specific technological aids, AIs are able to assess the intensity of exercises and suggest a change in it if it is inappropriate for the user’s needs; and (iv) Injury prevention and modification of exercises, based on the current health status of the exercising athlete [53,54,55,56,57,58,59].

Benefits and Challenges:-Improved diagnosis and treatment of diseases;-Personalization of medicine and treatment for individual patients;-Reduced workload for health care providers and improved efficiency;-Potential for discovering new therapeutic approaches and new drugs;-Use of AI can help identify and prevent health problems before they become serious;-Improved communication and collaboration among different members of the healthcare team;-Accurate monitoring of physical activities and health parameters;-Providing personalized feedback and artificial intelligence-based advice to improve fitness and health;-Increased motivation and adherence to physical activity;-Prevention and management of lifestyle-related chronic diseases;-AI can help identify behavior patterns and provide early interventions to improve health and well-being;-The use of wearable devices and AI-based fitness systems can encourage a healthier and more active lifestyle;-Integration of AI into existing clinical practice and interoperability with healthcare systems;-Managing large volumes of data and ensuring data quality;-Ensuring fairness and reducing algorithmic bias in the treatment of patients;-Difficulties in ensuring the adoption and acceptance of AI by patients and health professionals;-Improving human-machine interaction to make systems more intuitive and user-friendly;-Personalize fitness systems to suit individual needs and preferences;-Maintain long-term user interest and engagement;-Scalability and adaptability of AI-based fitness systems to different populations and contexts.

### 2.3. Virtual Reality (VR) and Augmented Reality (AR)

Virtual reality and augmented reality are transforming the way people exercise and interact with the tele-exercise environment. VR offers fully immersive training experiences, allowing users to exercise in virtual environments that can range from realistic settings to fantastical worlds. AR, on the other hand, overlays digital elements in the real environment, enhancing the training experience with additional information, such as performance statistics, tips, and personalized goals [59,60].

For example, Farrow et al. explore the application of virtual and augmented reality in sports coaching and skill acquisition, with a particular focus on rugby. The authors show how VR and AR can improve athletes’ learning and motivation by providing more immersive and personalized coaching experiences [61].

Benefits and Challenges:-Provide a safe and controlled training environment for learning and perfecting skills;-Enhance learning through repetition and focused practice;-Improve performance analysis and feedback to athletes;-Increase athlete engagement and motivation during training;-Virtual reality can help develop mental and cognitive skills, such as decision-making and situational awareness;-The use of virtual reality can facilitate collaboration and training among athletes and coaches from different geographical locations;-Addressing technical limitations, such as latency and graphics resolution;-Ensuring the validity and transferability of skills learned in virtual reality to the real world;-Reduce costs and increase the accessibility of virtual reality technology;-Adapting and customizing virtual reality systems for different sports and skill levels;-Managing potential side effects, such as motion sickness, while using virtual reality.

Kim and Park explore the potential of virtual and augmented reality in exercise and sports psychology research. The authors discuss how these technologies can enhance the training experience, increase user motivation and satisfaction, and offer new research opportunities in the field of exercise psychology. The challenges and opportunities for the adoption of these technologies in clinical and sports practice are also discussed [62].

### 2.4. Blockchain and Data Security

Blockchain technology can be used in tele-exercise to ensure the security and privacy of user data. Because personal information, such as health and performance data, is extremely sensitive, blockchain can ensure that such data is stored and shared securely and transparently. In addition, blockchain can facilitate the creation of incentive and reward systems to motivate users to achieve their fitness goals. Casino et al. examine blockchain-based applications through a systematic literature review and present the classification and open issues related to these applications. The article describes how blockchain technology can be used in tele-exercise to ensure the security and privacy of user data and facilitate the creation of incentive systems [63].

Benefits and Challenges:-Improve the data security of users;-Ensure privacy protection and comply with relevant regulations;-Making registration procedures easier and more secure for users.

### 2.5. Mobile Applications and Tele-Exercise Platforms

Mobile apps and tele-exercise platforms are becoming increasingly popular due to their ease of use and the ability to access a wide range of services and features. These platforms offer users the ability to take online fitness classes, track their progress, share results with friends, and even compete in virtual challenges. In addition, tele-exercise platforms can be customized to users’ needs and preferences, enabling a tailored workout experience. Silva et al. carried out a systematic review and meta-analysis of previous studies to evaluate the effectiveness of mobile applications running on smartphones in promoting physical activity. Twenty-one randomized controlled trials using different mobile applications were included. The meta-analysis showed that the use of mobile applications running on smartphones can lead to significant increases in physical activity. The authors highlighted the importance of customized mobile applications to improve the effectiveness of these technologies and the need to continuously evaluate the effects of interventions. The authors concluded that mobile applications running on smartphones can be effective in promoting physical activity, but further research is needed to identify the characteristics of the most effective interventions and to better understand how to customize them to fit individual needs [64]. In addition, the use of handheld devices would increase interest in physical activity among the younger segments of the population, who are usually more accustomed to the use of this technology [65]. This does not mean that portable devices are the exclusive preserve of children and adolescents; in fact, early preliminary studies show that the adult population also receive the benefits described above [66].

Benefits and Challenges:-Improvement of exercise-related habits;-Reducing sedentary lifestyle and its negative effects;-Facilitating access to tele-exercise programs for different types of users;-Increase participants within a young age range;-Increase the customization of exercise programs;-Ensure optimal performance of the various proposed tele-exercise programs.

### 2.6. Wearable Technology

Wearable technology is revolutionizing the field of tele-exercise, giving users the ability to constantly monitor their health and performance while exercising. As already described in the IoT section, devices such as smartwatches, fitness bracelets, and smart clothing can collect a wide range of data, including heart rate, oxygen consumption, posture, and sleep quality. This information can be analyzed and used to provide personalized feedback and suggestions, improving workout effectiveness and the user’s overall health.. In Kruse et al.’s systematic review, the authors examine the role of telemedicine in patient satisfaction and discuss the potential of telemedicine in providing remote support and counselling to telehealth users. The article highlights how telemedicine can improve access to personalized counselling and coaching services, reducing geographic barriers and contributing to a more engaging and satisfying user experience [67]. Dialogue between different devices is a key resource that tele-exercise must exploit to be optimal. Within this framework, wearable devices interface with all that instrumentation used by the user that can communicate with them. Monitoring the subject’s physical condition through worn devices sends the signal to the instrumentation used by the user himself by changing parameters such as resistance, speed, and lengths in a manner specific to the instrument being used and in real time.

Benefits and Challenges:-Simplicity and immediacy in device use;-Increase in user satisfaction levels;-Monitoring the physical condition of participants in tele-exercise sessions;-Improve appeal to older age groups.

### 2.7. Big Data and Predictive Analysis

Big data analytics is becoming increasingly important in the field of tele-exercise, as it enables the extraction of valuable information from the large volumes of data collected by IoT devices and fitness platforms. This information can be used to identify patterns and trends that can help predict and prevent injuries, improve performance, and promote long-term health [68]. In addition, predictive analytics can be used to further customize training plans and fitness strategies based on users’ individual needs and goals. For example, predictive analytics can identify injury risk based on specific movement patterns or training loads, allowing coaches and users to modify their routines to minimize risk and improve safety. Li and Li present a deep learning-based artificial intelligence system for generating a guidance model for sports education and physical fitness assessment. The study demonstrates the effectiveness of the proposed system in predicting individuals’ physical performance and assessing physical fitness [69]. Another paper examines the applications of big data analytics and machine learning in physical activity monitoring, discussing the challenges and opportunities offered by these technologies. The authors discuss the challenges and opportunities presented by big data analytics in health and medical settings, such as managing and analyzing massive amounts of data from various sources, including electronic health records, wearable devices, medical imaging, and genome sequencing. The information provided by big data can lead to a better understanding of diseases, their causes and underlying molecular mechanisms, as well as the discovery of new treatments and personalized therapies. Of particular interest is the discussion regarding the use of big data analytics to improve the quality and efficiency of health care, reducing costs and improving disease monitoring and prevention. In addition, the article addresses ethical, legal, and privacy issues related to the use of big data in healthcare and provides some successful examples of big data analytics applications in the medical and healthcare sectors [70].

Benefits and predictive analysis:-Cross-utilization of available data to prevent the occurrence of health risks;-Data analysis for the implementation of individual tele-exercise programs;-Monitoring the behavior of individual user subjects;-Lack of currently developed predictive models;-Implementation of a specific analysis algorithm.

### 2.8. Social Fitness and Gamification

Tele-exercise is becoming increasingly social, with platforms and apps encouraging users to share their progress, participate in challenges and competitions, and interact with other fitness enthusiasts [71]. Gamification, or the application of game mechanics to fitness, is a powerful tool for increasing user motivation and engagement in tele-exercise [72] Badges, leaderboards, goals, and virtual rewards can be used to encourage users to constantly improve and reach new milestones in their fitness journey [73]. Cotton and Patel analyze the effects of gamification on several measures of physical activity adherence and outcomes, including motivation, adherence, frequency, and duration of physical activity. The authors also examine the different gamification elements used in fitness apps, such as challenges, rewards, leaderboards, and personalization. In general, the authors find that the use of gamification in fitness applications has positive effects on motivation and adherence to physical activity. However, the effects on frequency and duration of physical activity are less clear and may depend on the specific implementation of gamification. In addition, the authors stress the importance of considering users’ individual needs and preferences when designing gamification elements. For example, some users may prefer individual challenges, while others may prefer social challenges. In addition, the authors note that the effectiveness of gamification may also depend on the duration and frequency of the intervention and that further research is needed to better understand these factors [74].

Benefits and Challenges:-Use of rewards and gratification as stimuli to increase participation;-Finding a functional gamification methodology for all users;-Limit the possibility of course dropout/interruption as a result of defeats in the gamification activity.

### 2.9. Biotechnology and Advanced Health Monitoring

Biotechnology is opening up new possibilities in the field of tele-exercise, enabling advanced health monitoring and personalization of exercise experiences. Biotechnology devices and sensors, such as electromyographs and biosensors, can measure parameters such as muscle activity, stress level, and body composition [75,76]. This data can be used to tailor workouts to users’ specific needs and provide them with detailed information about their health and well-being. The analytical work conducted by Gülü et al. investigates levels of obesity, physical activity, and video game addiction among adolescents and studies how machine learning can be used to predict video game addiction.

The study examines how obesity, physical activity, and video game addiction are interconnected and how these relationships may influence adolescents’ physical and mental health. The authors use machine learning techniques to analyze the collected data and create predictive models that can identify risk factors and possible consequences of video game addiction. In addition, the study seeks to understand how different variables, such as age, gender, socioeconomic level, and other factors, may influence the relationship between obesity, physical activity, and video game addiction. The ultimate goal is to use this information to develop interventions and preventive strategies that can reduce the incidence of video game addiction and promote healthier behaviors among adolescents [41].

Benefits and Challenges:-360-degree, real-time analysis of the subject’s health condition;-Ability to adapt exercises to the practitioner’s state of health, in a manner directly dependent on the practitioner’s physical condition;-Correct setting of instruments if not performed by an experienced hand;-Facilitate their use by the user who is “unfamiliar” with technology.

### 2.10. Robotics and Automation

Robotics and automation are entering the field of tele-exercise, offering new possibilities for training and user support. Robots can be used as virtual personal trainers, providing instruction, feedback, and motivation during workouts. In addition, automation can be used to simplify and streamline the management of tele-exercise platforms, improving the user experience and enabling more personalized fitness experiences.

Some examples of robotic applications in tele-exercise include physical therapist robots that can guide users through specific rehabilitation exercises, and robotic assistive devices that can support users with disabilities or physical limitations during workouts [77] These devices can be programmed to adapt to individual needs and provide a personalized level of assistance [78]. In addition, automation can be used to improve the analysis of data collected by IoT devices and fitness platforms. For example, machine learning algorithms can be used to identify patterns in the data and generate personalized suggestions and recommendations for users. This can help improve the quality of workouts and maximize fitness results. Automation can also be used in the logistics and maintenance of tele-exercise equipment. For example, advanced automation systems can monitor the wear and tear and degradation of equipment and schedule preventive maintenance actions, reducing the risk of malfunctions and failures.

Robotics and automation are revolutionizing the tele-exercise industry, providing new opportunities for personalized training and user support, and improving the efficiency and safety of fitness platforms. As these technologies continue to evolve, their impact on tele-exercise is expected to become even more significant, helping to create more engaging and personalized fitness experiences. In the future, robotics and automation could open up new frontiers in tele-exercise, with the implementation of even more advanced and sophisticated technologies. For example, we could see the use of robots with advanced artificial intelligence that are able to learn and adapt to users’ needs and preferences, offering even more personalized and dynamic training programs. In addition, robotics and automation could help make tele-exercise more accessible to a wider audience, removing some of the economic and logistical barriers associated with working out in a gym or with an in-person personal trainer. For example, robotic devices could become cheaper and more accessible, allowing more people to enjoy the benefits of personalized fitness.

The increasing integration of technologies such as virtual and augmented reality, IoT and AI with robotics and automation could also lead to new formats of tele-exercise, such as immersive virtual training environments or gamified fitness experiences. This could make tele-exercise even more engaging and motivating, encouraging users to maintain an ongoing commitment to improving their health and well-being. However, it is also important to consider the challenges and ethical implications associated with the increasing use of robotics and automation in telehealth. For example, ensuring the security and privacy of user data and addressing concerns about automation and job losses in the fitness industry will be critical.

In conclusion, robotics and automation have the potential to radically transform the tele-exercise industry, offering new opportunities and challenges. As these technologies evolve and spread, it will be critical to consider both the benefits and implications associated with their use to ensure a sustainable and inclusive future in the fitness and wellness field [33].

Benefits and Challenges:-Implementation of robot instructors customized to the needs of the individual user;-Improve the quality of training and exercise activities;-Reduction of possible failure and malfunction of devices used in tele-exercise;-Decreasing manufacturing costs;-Avoid the possible reduction of jobs for “traditional” trainers;-Keeping users’ sensitive data safe.

## 3. Benefits of Tele-Exercise and New Emerging Technologies

### 3.1. Accurate and Personalized Performance Analysis and Health Monitoring

Personal medical data can be used to provide personalized feedback and targeted advice, helping users improve their workouts and lifestyle. In addition, continuous health monitoring can help identify any problems or worrying trends in a timely manner, enabling preventive or corrective interventions. Prakashan and colleagues review advanced techniques in wearable devices for point-of-care and their futuristic applications. The authors review recent technological advances in wearable devices for point-of-care care, which provide continuous, noninvasive monitoring of patient vital parameters, and discuss their possible applications in various medical fields. In particular, the authors explore the potential of these technologies in areas such as early diagnosis, prevention, personalized therapy, and remote patient management. The article aims to provide a comprehensive overview of wearable technologies for point-of-care, providing a critical analysis of their strengths and weaknesses and suggesting possible future developments [79].

### 3.2. Real-Time Feedback and Suggestions to Improve Training and Wellness

Emerging technologies in tele-exercise can offer real-time feedback and suggestions to help users improve their training and overall well-being. For example, artificial intelligence-based virtual coaches can analyze users’ performance data and provide suggestions on how to modify training for optimal results. In addition, tele-exercise platforms can use data analytics and machine learning algorithms to identify patterns and trends, helping users better understand their bodies and needs. Several artificial intelligence techniques can be used to improve the accuracy of fitness recommendations, such as filtered collaboration, content analysis, and machine learning [80].

### 3.3. Increased User Motivation and Engagement

Emerging technologies in tele-exercise can help increase user motivation and engagement, making exercise more fun and rewarding. Virtual and augmented reality, for example, can create immersive and engaging workout experiences that can help users stay motivated and adhere to their fitness programs. In addition, the integration of gamification elements, such as scores, leaderboards, and prizes, can encourage friendly competition and the achievement of personal goals. Virtual reality influences motivation, affect, pleasure and engagement during exercise. Several studies examine the impact of using virtual reality on perceptions of physical training and exercise outcomes. In particular, virtual reality can be used to improve user motivation to exercise, increase engagement and participation in physical activity, and improve user enjoyment and satisfaction in physical training. Thus, virtual reality could be an effective option for improving the physical training experience, but further studies are needed to determine its long-term effectiveness [81].

### 3.4. Opportunities for Early Identification of Health Problems and Their Prevention

By collecting and analyzing health data, emerging technologies in tele-exercise can help identify and prevent health problems early on. For example, continuous monitoring of vital parameters and health conditions can detect abnormalities or changes that could indicate ongoing or future health problems. This can enable users to intervene early and, in some cases, prevent the development of chronic diseases or debilitating conditions [82].

Recently, a new method the detection of aerobic metabolic threshold was proposed by some authors [83,84], relying only on heart-rate time series overcoming currently used visual inspection, and in the future, once validated for different activities, can be easily implemented in applications acquiring data from portable heart-rate monitors and during specific tele-exercise sessions. The implication is that fitness levels can be assessed automatically and personally accessible; this determination method can be used to plan and monitor training in athletes to improve their performance and in several pathologies as obesity [42] and cardiac patients [68,85].

Moreover, as introduced in par 3.2, tele-exercise can permit the collection of big data, and automatic procedures are necessary. The most recent strategy is to apply cutting-edge machine learning techniques. They consist of demanding from an algorithm the selection of cases by using a vast quantity of data. These machine learning classifiers offer an effective method for examining variations in functional networks [34,80].

## 4. Challenges and Issues in Tele-Exercise and Emerging Technologies

### 4.1. Privacy and Data Security Concerns

One of the main challenges associated with the adoption of emerging technologies in tele-exercise is ensuring the privacy and security of sensitive user data. The collection and processing of data on health, exercise habits, and other personal information can raise concerns about data protection and the risk of unauthorized access. Companies offering tele-exercise services must implement robust security measures, such as data encryption and multifactor authentication, to protect user information and ensure compliance with data protection regulations. Another challenge in tele-exercise is ensuring the reliability and accuracy of data collected by IoT devices and fitness platforms. The quality of data can vary depending on the device, manufacturer, and conditions of use, and the accuracy of measurements is critical to providing valuable feedback and advice. To address this challenge, it is important to promote quality and compliance standards for tele-exercise devices and platforms, ensuring that the information collected and analyzed is accurate and reliable [86].

### 4.2. Disparities in Technology Availability

The adoption of smart devices and tele-exercise services may be limited by economic, geographic, and social factors, creating potential inequalities in the availability and use of these technologies. For example, individuals with lower incomes may not be able to afford expensive IoT devices or subscriptions to premium tele-exercise platforms [87]. In addition, people living in rural or remote areas may have limited access to tele-exercise services due to poor Internet connectivity or lack of technological infrastructure [35,88]. To address these inequalities, it is important to promote accessibility and inclusion in the tele-exercise sector, such as by promoting low-cost solutions, creating subsidy and funding programs for disadvantaged populations, and developing technology infrastructure in underserved areas.

Richardson and colleagues present guidelines and tools to promote digital equity, or the equitable access to and effective use of technology and digital resources by all people, regardless of social, economic, or geographic background. The authors analyze the nature and importance of digital equity and propose a number of strategies and tools to address inequalities in the digital domain. These tools include training and digital literacy programs, targeted public policies, funding programs, and public–private partnerships. Their work emphasizes the importance of promoting digital equity to improve access to social, economic, and learning opportunities [89].

### 4.3. Overcoming Cultural Barriers and Distrust of Emerging Technologies

Another challenge in promoting the adoption of emerging technologies in tele-exercise is overcoming adoption resistance and cultural barriers. Some individuals may be skeptical about the usefulness of new technologies or concerned about the possible loss of human interaction and social support in tele-exercise. To overcome these barriers, it is important to promote awareness and education about the potential positive implications of emerging technologies in tele-exercise, such as improved health and well-being, access to a greater variety of exercise options, and the ability to connect with supportive virtual communities [90]. In addition, tele-exercise platforms and companies offering IoT devices should seek to integrate elements of human interaction and social support into their services, such as through the creation of online support groups, the ability to share progress and goals with friends and family, and the provision of human coaching in conjunction with AI-based training. The use of tele-exercise could be critical in combating sedentary behaviors in children and adolescents with obesity. Calcaterra et al. examine the impact of COVID-19 on physical activity levels and the consequent need for alternative solutions such as tele-exercise. Different approaches to tele-exercise are examined, including remote exercise programs, fitness apps, and active gaming platforms. The authors emphasize the importance of tailoring exercises to individual needs and providing motivational support to improve remote exercise adherence. In addition, the authors examine the effects of tele-exercise on health outcomes, such as reducing obesity and improving cardiovascular health. A difficult target audience, such as children, and their families, welcomed the use of these new possibilities offered by modern technology [40].

## 5. Opportunities for Future Research

New trends and technologies in telehealth offer numerous opportunities for future research, including:-Development of more advanced and personalized AI algorithms to analyze data collected from IoT devices and provide user-specific feedback and recommendations;-Integration of smart devices with other emerging technologies, such as virtual and augmented reality, to create more immersive and immersive workout experiences;-Studies on the long-term impact of smart devices and tele-exercise use on users’ health and well-being, as well as the prevention and treatment of chronic diseases;-Investigation of barriers to the adoption of smart devices and tele-exercise services, with the goal of improving access and equity in the use of these technologies.

Future research could focus on developing more advanced and personalized artificial intelligence algorithms that can more accurately analyze data collected from IoT devices. These algorithms could be used to provide specific feedback and advice to users based on their individual needs and preferences, thereby improving the effectiveness and efficiency of training. In addition, it could be interesting to explore how AI algorithms can adapt to users’ different abilities and health conditions to provide personalized and appropriate workout plans. For example, the research could focus on how to develop AI algorithms that take into account the needs of users with disabilities, chronic medical conditions, or recovering from injuries.

Another promising area of research involves integrating IoT devices with other emerging technologies, such as virtual reality and augmented reality, to create more immersive and immersive exercise experiences. Scholars could explore how to combine these different tools to provide a unique and personalized tele-exercise experience that takes into account the specific needs of each user. For example, the research could focus on how to use virtual reality to simulate realistic and challenging exercise environments that encourage users to engage in more intense exercise and achieve their fitness goals. Similarly, studies could investigate how augmented reality could be used to provide visual instruction and real-time feedback during workouts, improving technique and reducing the risk of injury. Future research could also investigate the long-term impact of smart devices and tele-exercise use on users’ health and well-being. Longitudinal studies could examine how the use of these technologies affects the prevention and treatment of chronic diseases, such as obesity, diabetes, and cardiovascular disease. In addition, the research could explore how the use of smart devices and tele-exercise can contribute to users’ mental well-being, such as by assessing the impact of these technologies on stress reduction, promotion of quality sleep, and prevention of depression and anxiety.

### Investigation of Barriers to the Adoption of Smart Devices and Tele-Exercise Services

Future studies could investigate barriers to the adoption of smart devices and tele-exercise services, with the goal of improving access and equity in the use of these technologies. These barriers could include economic, geographic, social, and cultural factors that influence the availability and use of smart devices and tele-exercise platforms.

Economic factors: Research could examine how the cost of smart devices and telehealth services may limit access for some segments of the population. Studies could identify strategies to reduce costs and make these technologies more accessible to a wider audience, such as through subsidies, financing, or innovative business models.

Geographic factors: Future studies could investigate whether access to telehealth technologies is limited by geographic factors, such as the availability of high-speed Internet connections or the presence of adequate infrastructure. Research could explore solutions to improve access in rural or developing areas, such as through the use of alternative communication technologies or upgrading existing infrastructure.

Social and cultural factors: Research could also examine how social and cultural factors influence the adoption and use of smart devices and tele-exercise services. For example, studies could investigate whether the use of these technologies is influenced by cultural norms, attitudes toward physical activity, or privacy and data security concerns.

Strategies to overcome any barriers: Future studies could identify and evaluate strategies to overcome various barriers to the adoption of smart devices and telehealth services. This could include training and awareness programs for users, partnerships between the public and private sectors to develop more affordable infrastructure and services, and the adoption of stricter security and privacy standards to protect user data and promote trust in the tele-exercise industry. Emerging technologies in tele-exercise thus offer numerous opportunities for future research that can help improve the accessibility, effectiveness, and personalization of fitness experiences. Deepening the understanding of these technologies and their potential applications can have a significant impact on the health and well-being of a wide range of users around the world.

## 6. International Perspectives and Future Solutions

The adoption of smart health and wellness devices applied to tele-exercise has the potential to transform the approach to health and wellness globally. To maximize the positive impacts of these technologies and ensure that the benefits are shared equitably, it is important to consider the following future perspectives and solutions. International cooperation among governments, nongovernmental organizations, research institutions and private sector companies can play a crucial role in accelerating the innovation and adoption of tele-exercise technologies globally. Some approaches to promoting international cooperation include:-Sharing best practices and global standards. This can help ensure that these technologies are used effectively and safely and that the benefits are shared more equitably globally;-Development of partnerships and collaborations. These partnerships can foster knowledge exchange, access to financial and technological resources, and the creation of support networks for the implementation of tele-exercise projects in different regions of the world;-Promotion of innovations through research and development. International cooperation can stimulate research and development of new technologies, for example, by jointly funding research projects, establishing centers of excellence, and promoting training and exchange initiatives among researchers and practitioners in the field.

Furthermore, to ensure that the benefits of tele-exercise technologies are shared equitably globally, it is critical to address the challenges related to accessibility and inclusion. It is important to reduce the costs of smart devices and associated services. This can be achieved through innovation in design and manufacturing, as well as through the development of innovative business models that reduce costs for end users, for example, by offering free or low-cost services or using subscription systems. Another key step is the improvement and expansion of communications infrastructure, such as broadband coverage and mobile services. Governments and international organizations can work together to promote investment in communications infrastructure, particularly in rural and remote areas where access to such services may be limited. To ensure that people of all ages and educational levels can effectively use tele-exercise technologies, it is important to promote digital literacy and provide training opportunities. Products and services need to be developed that take into account diverse cultural, linguistic, and enabling needs. This may include designing devices and applications that support different languages, customizing training experiences according to cultural preferences, and creating accessible features for users with disabilities.

Research and innovation should be supported and funded to improve the quality and effectiveness of tele-exercise technologies. Some key areas of interest include:-Development of more advanced and accurate devices and sensors. They can improve the quality and reliability of data collected by smart devices, enabling better personalization of tele-exercise experiences and more accurate monitoring of users’ health and well-being;-Developments in device and platform integration can enable users to easily access and use data collected by smart devices across different platforms and tele-exercise services. This can improve the user experience and ensure that health and wellness information is easily accessible and usable;-Trials on the effectiveness and impact on individual health and well-being are critical to understanding how these technologies can best be used to promote global health and well-being. Studies should examine the short- and long-term impact of the use of smart devices and tele-exercise services on users’ physical and mental health, as well as their habits and behaviors. In addition, research should investigate the effectiveness of these technologies in preventing and treating chronic diseases and lifestyle-related health conditions.

The next step would be to promote the integration of tele-exercise practice within national health care systems. Governments and private sector companies should work together to integrate these technologies into national healthcare systems. This collaboration could include creating tax incentives or funding for companies that develop smart devices and tele-exercise services, as well as promoting pilot projects that demonstrate the effectiveness of these technologies in preventing and treating chronic diseases and promoting health and well-being. It is important to train healthcare professionals to effectively use these devices and services in the clinical practice setting. These training programs should cover aspects such as interpreting data collected by smart devices, identifying the best data-driven intervention strategies, and interacting with patients to promote the adoption of telehealth technologies. To promote the responsible adoption of tele-exercise technologies it is important to develop international standards and guidelines that cover aspects such as the quality and security of smart devices, privacy and data protection, and accessibility and equity in the use of these technologies.

In Figure 2 a comprehensive overview of the use of new devices in tele-exercise is shown:

## 7. The Ideal Tele-Exercise App

The comprehensive overview carried out enabled us to define what the characteristics of the ideal tele-exercise application that should be developed in the future might be. In fact, many sections analyzed so far have overlaps and interconnections. All of this is essential for understanding and analyzing the individual sections, but dispersive in achieving the idea of an optimal app.

The reality of the modern world confronts us with extensive technological and digital interconnectedness. Within this context, a Tele-exercise App must be able to interface and communicate with different devices. It must have the ability to communicate with wearable devices that monitor the subject’s health condition in real time and with the databases of different national health systems, from which it will have to obtain the user’s data. All of this information will need to be integrated in order to return to the user the exercise best suited to his or her needs at that specific time and for the goal set in an equally specific manner. Having access to the relevant set of personal information raises for us the consideration that the App will have to be able to ensure optimal privacy management that complies with the laws of the different countries in which it is used.

In addition, nowadays, we find ourselves within a historical period where we are all able to easily switch from one device to another, and therefore the Tele-exercise App will have to be able to be executable on different platforms while maintaining the same degree of efficiency. To ensure maximum uptake, its user interface will need to be intuitive and customizable to enable individuals of different ages and different technological backgrounds to achieve the best performance.

The presence of features such as the ability to implement augmented/virtual reality sessions, or competitive/gaming sessions, are not essential, but would allow for an increase in absorption, gratification, positive stimuli, and incentives in the long run. To date, these features may seem excessive (especially when considered as a whole) on a single app compared to the development of more specific, industry-specific applications. Fortunately, scientific advances and the steady increase in digital literacy give hope that this is not such a remote possibility. In the future, most likely, the physical well-being of the general population will come through digital and telematics support that will complement and interface with current ways.

## 8. Conclusions

Regular physical activity brings numerous benefits in the prevention of diseases. Its importance has been attested in several fields concerning health, especially in the prevention of obesity and cardiovascular diseases [1,10,36,91]. Tele-exercise has become increasingly popular in recent years due to rapid advances in digital technology and a growing awareness of the need to maintain an active lifestyle for health and well-being. This has led to an expansion of available tele-exercise options, which include portable devices such as fitness trackers, fitness, and health apps, virtual coaching programs, and more [6,14,51].

One of the emerging trends in tele-exercise is the use of smart devices, such as smartwatches and virtual assistants. These devices can provide greater interactivity and personalization in monitoring physical activity, nutrition, and overall health. In addition, the use of smart devices can increase user motivation through gamification and personalized goal setting. The integration of emerging technologies such as IoT and AI into tele-exercise is leading to an era of personalized and interconnected solutions that can improve quality of life for people. For example, the use of wearable sensors and IoT devices can provide real-time data on physical activities, heart rate, and other health parameters, enabling more accurate monitoring and better personalization of tele-exercise solutions [52,64,74]. In addition, AI can be used to analyze the collected data and provide personalized recommendations to improve health and well-being. However, as the use of digital technologies in tele-exercise increases, significant privacy and data security challenges arise. It is critical that personal information collected from users is protected from unauthorized access and that data is managed responsibly and in accordance with privacy laws. In addition, equitable access to tele-exercise technologies is an important concern, as a digital divide may exist between those who can afford devices and applications and those who cannot [92].

The integration of tele-exercise with medical research and practice is another major challenge. While tele-exercise can be used as part of a comprehensive interdisciplinary approach to health and wellness, it is important that tele-exercise solutions are evidence-based and effectively integrated with existing medical practice. This requires a collaborative approach among experts in medicine, technology, and research to develop tele-exercise solutions that are evidence-based and adaptable to individual patient needs. In addition, access to data generated by tele-exercise devices can provide valuable information for medical research. For example, data on physical activity, sleep, and nutrition patterns can be used to study correlations between lifestyle and health. However, it is important to ensure these data are collected and managed responsibly and user privacy is guaranteed [93].

To address these challenges, it is essential that there is a concerted effort by governments, international organizations, industries, and institutions to promote collaboration, development of global policies and standards, investment in research, and technological innovation. In particular, collaboration between fields such as medicine, engineering, psychology, and sociology can provide a broader view of tele-exercise and its impacts on health and well-being.

In addition, there are many opportunities for future research in the field of tele-exercise. For example, studies can explore the effectiveness of tele-exercise solutions in different population groups, including children, the elderly, and people with disabilities. In addition, research can focus on developing tele-exercise solutions that are culturally appropriate and adaptable to the diverse needs of different populations. Monitoring progress and achievements internationally will be critical to ensure that the benefits of tele-exercise are shared equitably and that innovations are implemented effectively and responsibly [59,68,75].

Tele-exercise represents a significant opportunity to improve the quality of life for people around the world [44]. However, to fully exploit the potential of these technologies, it is critical to address challenges related to data privacy and security, equitable access to technologies, integration with medical research and practice, and promoting collaboration across sectors. Future research can focus on identifying new opportunities and exploring potential synergies between telehealth and other emerging disciplines and technologies. In addition, research can focus on analyzing the results obtained so far and evaluating the effectiveness of tele-exercise solutions in different contexts. For example, studies can evaluate the impact of tele-exercise solutions on reducing chronic disease risk, weight management, stress reduction, and mental health promotion.

Furthermore, it is critical to emphasize the importance of ensuring that everyone has access to tele-exercise technologies. Tele-exercise solutions may be inaccessible to those who cannot afford the devices and applications. This digital divide may lead to health and wellness disparities among different populations. To address this challenge, organizations and governments can develop policies and programs to provide access to all members of society.

## Figures and Tables

**Figure 1 healthcare-11-01805-f001:**
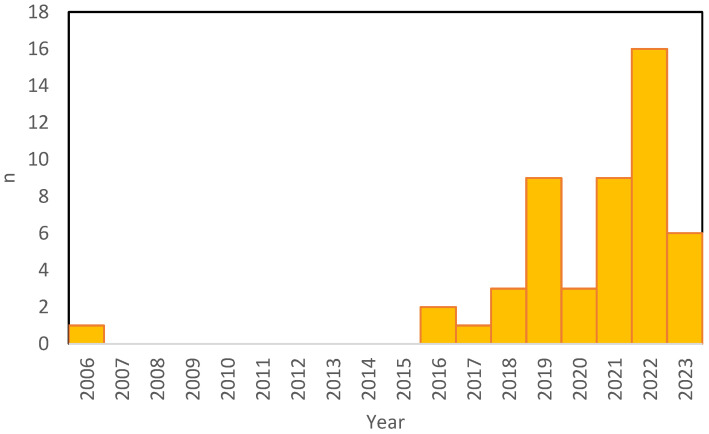
Number of papers found using the keyword “Tele-exercise” on PubMed.

**Figure 2 healthcare-11-01805-f002:**
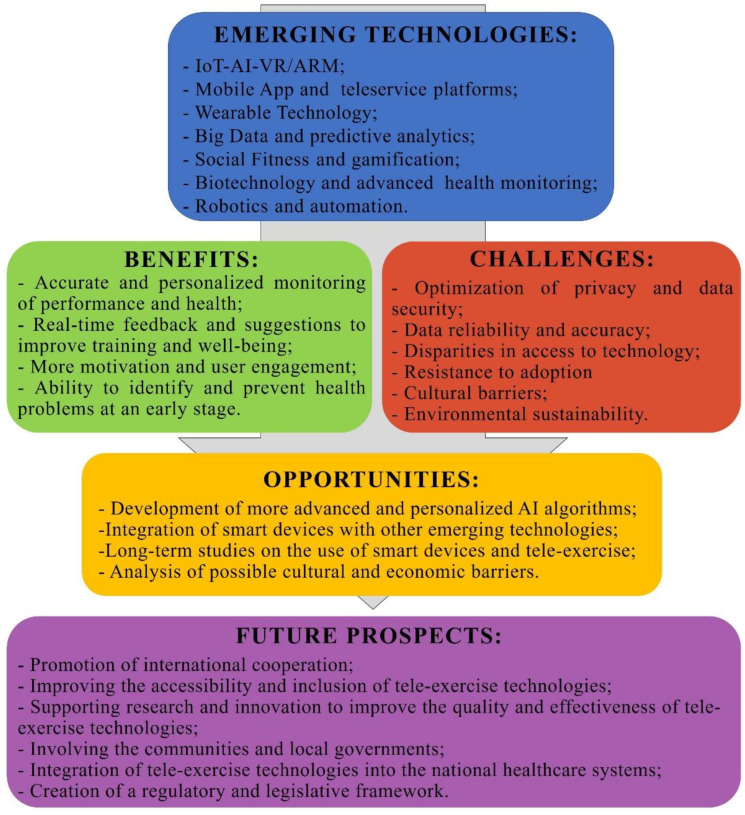
Smart devices for health and wellness applied to tele-exercise.

**Table 1 healthcare-11-01805-t001:** PUB-MED results from 2021 to 2023.

Keyword	AND	Results	Ref. Number
Tele-exercise		29	[4,5,6,8,9,11,15,16,17,18,19,20,21,22,23,24,25,26,27,28,29,30,31,32,33,34,35,36]
	COVID	13	[4,5,6,9,29,32,37,38,39,40]
	Obesity	6	[30,37,39,40,41,42]
	Children	4	[37,39,43,44]
	Cancer	2	[15,22]
	SCI	3	[5,17,45]

## Data Availability

Not applicable.

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
