# Peer review of "Smart Devices for Health and Wellness Applied to Tele-Exercise: An Overview of New Trends and Technologies Such as IoT and AI"

_healthcare, 2023, doi:10.3390/healthcare11121805_

Round 1

Reviewer 1 Report

This article highlights the numerous benefits of regular physical activity and the rising popularity of tele-exercise, enabled by digital technology. The integration of smart devices and emerging technologies like IoT and AI is revolutionizing the way we approach fitness and health. These advancements allow for greater interactivity, personalization, and motivation through gamification and goal setting.

However, the review also raises some concerns that need to be addressed. One significant concern is the issue of privacy and data security. As the use of digital technologies in tele-exercise increases, it becomes crucial to protect personal information and ensure responsible data management in accordance with privacy laws.

Equitable access to tele-exercise technologies is another important consideration. There is a digital divide between those who can afford these devices and applications and those who cannot. To bridge this gap, efforts should be made to ensure that tele-exercise is accessible to all members of society.

Integrating tele-exercise with medical research and practice is a major challenge as well. It is essential to develop evidence-based solutions and effectively integrate tele-exercise into existing medical practices through interdisciplinary collaboration. Additionally, responsibly collecting and managing data generated by tele-exercise devices can provide valuable information for medical research, but user privacy must be guaranteed.

To overcome these challenges, a concerted effort is needed from governments, international organizations, industries, and institutions. Collaboration, the development of global policies and standards, research investment, and technological innovation are essential. Interdisciplinary collaboration, especially between fields like medicine, engineering, psychology, and sociology, can provide a broader understanding of tele-exercise's impacts on health and well-being.

The article concludes by emphasizing the need for future research to explore the effectiveness of tele-exercise solutions in different population groups and cultural contexts. Monitoring progress internationally and evaluating the outcomes of tele-exercise interventions in areas such as chronic disease risk reduction, weight management, stress reduction, and mental health promotion are important avenues for future exploration.

Overall, tele-exercise presents an exciting opportunity to enhance the quality of life worldwide. However, addressing concerns related to data privacy, equitable access, integration with medical research, and fostering collaboration across sectors are crucial for maximizing the potential benefits of tele-exercise technologies.

No issue.

Author Response

Thanks to all the reviewers.

Your considerations have been essential in improving the quality of our work. They are precise and accurate, and we hope we have fully satisfied you.

You will find all changes entered via review mode in order to make them easier to identify. 

We thank you for the complete and perfect analysis of our article. We especially appreciate that the message was fully caught. We also hope that the additional paragraphs, edits and additions will reinforce the idea that was already made regarding our work.

English has been corrected where necessary, and the style improved.

Reviewer 2 Report

This review article discusses the new trends of IoT and AI on smart devices for health and wellness applied to tele-exercise. However, the references are not enough for a review article, and most of them are not relatively new since AI technology iterates very fast. The reviewer recommends the authors discuss more recent AI research on health and wellness applications, such as CVPR 2023. In addition, more figures or tables are needed to present these applications more clearly and intuitively.

Minor editing of English language required

Author Response

Thanks to all the reviewers.

Your considerations have been essential in improving the quality of our work. They are precise and accurate, and we hope we have fully satisfied you.

You will find all changes entered via review mode in order to make them easier to identify.

A: Unfortunately, at the time of writing it is not possible for us to mention Computer Vision and Pattern Recognition 2023, as it is not yet available.

Aware of the speed at which discoveries and developments in the field of artificial intelligence are occurring, we have tried to supplement our work through "non-classical" sources.

In fact, the information found on online sites, which are much more easily updated than scientific papers, are certainly more recent and we hope are sufficient to express our views on AI and exercise.

As suggested, we increased the number of figures, inserted a table, and reshaped the present figure.

In the 7th section, we try to define the ideal characteristics of a tele-exercise app. We added this new section thanks to your suggestion to propose our mandatory paths for the future research and improvements in this field.

We believe that now the ideology that prompted us to create this review is more understandable, and we thank you for the suggestions sent.

Reviewer 3 Report

The review article aims at an overview on different emerging technologies and trends on tele-exercise applications. The emerging technologies are internet of things, artificial intelligence, virtual reality and augmented reality, blockchain and data security, mobile applications, wearable technology, big data and predictive analysis, social fitness and gamification, biotechnology and advanced health monitoring and robotics and automation. A review article normally provides the readers with an in depth understanding of a field and highlights key gaps and challenges to be addressed in future research. I don’t see such a contribution in this article. Basically, it reads like a review article of the review articles for each of the mentioned emerging technologies. The contents are totally superficial. The specific requirements of tele-exercise applications and how they differ from other applications are not discussed. Most of the listed benefits and challenges of the technological developments are independent of tele-exercise applications. Some parts don’t have any specific relation to tele-exercise at all (e.g. chapter 4.4)

Parts of the contents are repeated several times: The sentence (line 407-409) “Devices such as smart watches, activity trackers, and smart scales make it possible to collect detailed data on physical activities, heart rate, calorie consumption, sleep quality, and other parameters” appeared very similar also in chapter 2.1 and 2.6. Chapters 2.2 on artificial intelligence and 2.7 on big data also have an overlap.

There are hardly any useful hints on the things that should be addressed by future research. The mentioned research opportunities like to develop “more advanced and personalized AI algorithms” (Line 543) or “more advanced and accurate devices and sensors” (Line 658) are just useless hints. Most of the mentioned research challenges are obvious in the technological fields and don’t have a specific relation to tele-exercise applications.

Figure 1 is hardly readable.

Author Response

Thanks to all the reviewers.

Your considerations have been essential in improving the quality of our work. They are precise and accurate, and we hope we have fully satisfied you.

You will find all changes entered via review mode in order to make them easier to identify.

The review article aims at an overview on different emerging technologies and trends on tele-exercise applications. The emerging technologies are internet of things, artificial intelligence, virtual reality and augmented reality, blockchain and data security, mobile applications, wearable technology, big data and predictive analysis, social fitness and gamification, biotechnology and advanced health monitoring and robotics and automation.

 A review article normally provides the readers with an in depth understanding of a field and highlights key gaps and challenges to be addressed in future research. I don’t see such a contribution in this article.

A: new recent papers are considered to deeply explain key gaps and challenges see [5, 6, 9, 15-26, 28-36, 39-41, 45-52, 57, 58, 63-65, 70, 71, 83-85, 87]

A Pub-MED research is update and represented inside the label to facilitate the overview of actual literature.

At the same time, we integrated a new figure about the same data to give an immediately perspective to the readers. .

Basically, it reads like a review article of the review articles for each of the mentioned emerging technologies. The contents are totally superficial. The specific requirements of tele-exercise applications and how they differ from other applications are not discussed. Most of the listed benefits and challenges of the technological developments are independent of tele-exercise applications. Some parts don’t have any specific relation to tele-exercise at all (e.g. chapter 4.4)

Parts of the contents are repeated several times: The sentence (line 407-409) “Devices such as smart watches, activity trackers, and smart scales make it possible to collect detailed data on physical activities, heart rate, calorie consumption, sleep quality, and other parameters” appeared very similar also in chapter 2.1 and 2.6. Chapters 2.2 on artificial intelligence and 2.7 on big data also have an overlap.

A: As suggested, we have selected the benefits only related to tele-exercise. Chap 4.4 has been removed and its content has been redistributed in other sections. The content overlap between Sections 2.1 and 2.6 is there but it is consequential that it is so. In fact, wearable devices are part of IoT, so we have tried to add an explanation within section 2.6 to clarify the concept. The same argument can be made regarding the overlap between sections 2.2 and 2.7. It is not automatic that data collected and analyzed by AIs are then placed within a BIG DATA system. Rather, this communication, in our view, is essential to achieve optimal output in rendering the tele-exercise for the user.

At beginning of Par 3.1 a sentence is removed. Chap 6 some repetitions are removed (see pag 14) In chap 2Benefit and Challenges are summarized

There are hardly any useful hints on the things that should be addressed by future research. The mentioned research opportunities like to develop “more advanced and personalized AI algorithms” (Line 543) or “more advanced and accurate devices and sensors” (Line 658) are just useless hints. Most of the mentioned research challenges are obvious in the technological fields and don’t have a specific relation to tele-exercise applications.

A: We understand that within such a broad and complex topic the inclusion, in a peripheral and non-detailed way, of an issue such as environmental sustainability may be superfluous. It probably should be explored in greater depth to give it the proper weight so we accept the suggestion to remove it in order to facilitate reading.

Correctly it has been pointed out to us that such an examination, without a paragraph devoted to the specifics of tele-exercise, may give the feeling that the article has the potential to be applied to any new technology.

In this regard, we have included a concluding paragraph that is able to provide our vision of the IDEAL app based on the above.

Round 2

Reviewer 2 Report

The authors have improved the manuscript according to the reviews.